# Exceedingly High Performance Top-Gate P-Type SnO Thin Film Transistor with a Nanometer Scale Channel Layer

**DOI:** 10.3390/nano11010092

**Published:** 2021-01-03

**Authors:** Te Jui Yen, Albert Chin, Vladimir Gritsenko

**Affiliations:** 1Department of Electronics Engineering, National Chiao Tung University, Hsinchu 300, Taiwan; yenrick42269.ee05g@g2.nctu.edu.tw; 2Rzhanov Institute of Semiconductor Physics, Siberian Branch, Russian Academy of Sciences, Novosibirsk 630090, Russia; grits@isp.nsc.ru; 3Novosibirsk State University, 2 Pirogov Street, Novosobirsk 630090, Russia; 4Novosibirsk State Technical University, 20 Marks Avenue, Novosibirsk 630073, Russia

**Keywords:** monolithic 3D, 3D brain-mimicking IC, SnO TFT

## Abstract

Implementing high-performance n- and p-type thin-film transistors (TFTs) for monolithic three-dimensional (3D) integrated circuit (IC) and low-DC-power display is crucial. To achieve these goals, a top-gate transistor is preferred to a conventional bottom-gate structure. However, achieving high-performance top-gate p-TFT with good hole field-effect mobility (*μ_FE_*) and large on-current/off-current (I_ON_/I_OFF_) is challenging. In this report, coplanar top-gate nanosheet SnO p-TFT with high *μ_FE_* of 4.4 cm^2^/Vs, large I_ON_/I_OFF_ of 1.2 × 10^5^, and sharp transistor’s turn-on subthreshold slopes (*SS*) of 526 mV/decade were achieved simultaneously. Secondary ion mass spectrometry analysis revealed that the excellent device integrity was strongly related to process temperature, because the HfO_2_/SnO interface and related *μ_FE_* were degraded by Sn and Hf inter-diffusion at an elevated temperature due to weak Sn–O bond enthalpy. Oxygen content during process is also crucial because the hole-conductive p-type SnO channel is oxidized into oxygen-rich n-type SnO_2_ to demote the device performance. The hole *μ_FE_*, I_ON_/I_OFF_, and *SS* values obtained in this study are the best-reported data to date for top-gate p-TFT device, thus facilitating the development of monolithic 3D ICs on the backend dielectric of IC chips.

## 1. Introduction

Metal-oxide Thin film transistors (TFTs) [1,2,3,4,5,6,7,8,9,10,11,12,13,14,15,16,17,18,19,20,21,22,23,24,25,26,27,28,29,30,31,32,33,34,35] have drawn considerable attention due to their high mobility, low fabrication temperature, and simple fabrication process, making them suitable for advanced display [1,2,3,4,5,6] and monolithic three-dimensional (3D) integrated circuit (IC) [15,16,17,18,19,20,21] on amorphous inter-metal-dielectric (IMD) of a Si chip. To reach low DC power consumption, both high performance n- and p-type TFTs are necessary to form the complementary metal-oxide-semiconductor (CMOS) logic. Although high-performance n-TFTs with high field-effect mobility (*μ_FE_*), sharp subthreshold swing (SS), and large on-current/off-current (I_ON_/I_OFF_) values [12,13,14,15] have been reported, achieving reasonable performance p-TFTs is much more challenging [16,17,36,37]. Moreover, a top-gate structure is more suitable than a conventional bottom-gate device for high integration density and easy fabrication [24,25,26]. Previously, we have reported the bottom-gate SnO p-TFT, which has higher *μ_FE_* than Cu_2_O p-TFT [16,17]. In the current study, we further used the SnO channel to fabricate top-gate coplanar nanosheet p-TFT. Because the gate insulator was deposited after the SnO layer, the deposition and post-annealing conditions are crucial to device performance. This is because the p-type SnO is highly sensitive to oxygen partial pressure (*O_pp_*) and annealing temperature, and can be easily oxidized into oxygen-rich Sn_2_O_3_, Sn_3_O_4_, or SnO_2_ [27,28,29]. Moreover, the weak Sn–O bond enthalpy [38] facilitates Sn diffusion into high-dielectric-constant (high-κ) HfO_2_ insulator at elevated temperature, thus degrading device performance. In this study, the above challenges were successfully overcome, and high-performance top-gate nanosheet SnO p-TFT was achieved with high *μ_FE_* of 4.4 cm^2^/Vs, large I_ON_/I_OFF_ of 1.2 × 10^5^, and sharp *SS* of 526 mV/decade, indicating a high potential for future monolithic 3D and brain-mimicking IC applications [15,17,18,19,20,21].

## 2. Materials and Methods

The coplanar top-gate nanosheet SnO p-TFTs were fabricated on the Si wafer with a 500-nm-thick SiO_2_ IMD layer on Si wafer. The 7-nm-thick nanosheet SnO layer was deposited through reactive sputtering with 50 W DC power from a Sn target under *O_pp_* values of 14.2%, 25%, and 33.3% ambient, respectively. All the SnO p-TFT samples were annealed under 200 °C in N_2_ ambient for 45 min. Next, 40-nm high work-function Ni was deposited using an e-gun evaporator for the Schottky-barrier source and drain electrodes [39,40]. Subsequently, 50-nm HfO_2_ gate dielectric was deposited by e-beam evaporation with a rate of 0.2 Å/sec. HfO_2_ post-annealing was performed in N_2_ ambient at 100 °C and 200 °C. Finally, 50-nm gate electrode Al was deposited and patterned. The transistor’s length and width were 50 and 400 μm, respectively. The current-voltage characteristics of top-gate SnO p-TFT were measured through the HP4155B parameter analyzer and a probe station. The field-effect mobility values (*μ_FE_*), subthreshold slope (*SS*) and on-current/off-current (I_ON_/I_OFF_) values were extracted at a standard and small V*_DS_* = −0.1 V. The cross-sectional image of device structure was obtained from FEI Talos F200X high-resolution transmission electron microscope (TEM). The surface roughness of HfO_2_ films were obtained via Atomic Force Microscope (AFM) using DIMENSION 3100. The X-ray photoelectron spectroscopy (XPS) analyses of HfO_2_ films and SnO films were executed by Thermo Nexsa. The secondary ion mass spectrometry (SIMS) depth profiles of Sn, Hf and O atoms were obtained by CAMECA IMS-6fE7.

## 3. Results

Figure 1a illustrates the top-view photograph of top-gate nanosheet SnO p-TFT, where the light-reflective Al metal-gate is on the top of the device. Figure 1b depicts the cross-sectional transmission electron microscope (TEM) image of the device structure with top Al-metal-gate, HfO_2_ gate-dielectric, and p-type channel SnO on SiO_2_ IMD. The thickness of Al, HfO_2_, and nanosheet SnO is 50, 50, and 7 nm, respectively.

The *O_pp_* is critical for top-gate nanosheet SnO p-TFT, where the SnO channel was made by sputtering from a metal Sn target under different *O_pp_* conditions. This is because the SnO can be oxidized into oxygen-rich SnO_2_ [16]. The *O_pp_* can be expressed as follows:(1)$$O_{pp}\;=\;\frac{P_{O_{2}}}{P_{O_{2}+}P_{Ar}}\;\times \;100\%,$$
where $P_{O_{2}}$ and $P_{Ar}$ are the pressures of *O*_2_ and *Ar* in a sputtering system, respectively. For comparison, the *O_pp_* values were adjusted to 14.2%, 25% and 33.3% during sputtering. Figure 2a,b show the drain-source current versus gate-source voltage (|*I_DS_*|-*V_GS_*) and *μ_FE_*-*V_GS_* characteristics of top-gate nanosheet SnO p-TFT devices, respectively, under different *O_pp_* values. The top-gate p-type SnO device exhibits the highest I_ON_ and the lowest leakage I_OFF_ at the 25% *O_pp_* condition. The device with the best I_ON_ and I_OFF_ is also consistent with the highest *μ_FE_*. The *μ_FE_* values were 1.5, 4.4 and 2.6 cm^2^/Vs at *O_pp_* of 14.2%, 25% and 33.3%, respectively. Here the *μ_FE_* values were obtained at the standard and a small V*_DS_* of −0.1 V. Such abnormal *μ_FE_* on *O_pp_* is ascribed to the following reasons. The device *μ_FE_* increases with the increase in *O_pp_* from 14.2% to 25% due to the increased oxygen content in SnO_x_, with x ≤ 1, and device performance degrades at a high *O_pp_* of 33.3% owing to the formation of oxygen-rich SnO_x_, with x > 1. Under high *O_pp_*, SnO_x_ becomes n-type electron-conductive SnO_2_ [12,13,14], which lowers the hole *μ_FE_* under negative *V_GS_*.

The device integrity in top-gate nanosheet SnO p-TFT is also dependent on HfO_2_ annealing temperature. To avoid plasma damage to the SnO channel layer, the high-κ HfO_2_ gate dielectric was deposited using an e-beam evaporator and subjected to post-annealing at 100 and 200 °C for 30 min under N_2_ ambient. Here the SnO layers were deposited under 25% *O_pp_* and annealed at 200 °C under N_2_ ambient. Subsequently, the HfO_2_ were deposited and annealed at 100 °C or 200 °C under the N_2_ ambient. The |*I_DS_*|-*V_GS_* and *μ_FE_*-*V_GS_* characteristics of SnO p-TFTs with 100 and 200 °C post-annealing are shown in Figure 3a, b, respectively. The I_ON_/I_OFF_ and *μ_FE_* values of the SnO p-TFT at 100 °C post-annealing are 1.2 × 10^5^ and 4.4 cm^2^/Vs, respectively, which are much better than those obtained at 200 °C: 4.6 × 10^2^ and 1.44 cm^2^/Vs, respectively. The I_ON_/I_OFF_ is even better than previous bottom-gate SnO p-TFT [16] possibly due to the thinner SnO channel used in this study, which slightly degrades the *μ_FE_*. A thin channel layer is needed to fully deplete the conductive oxide semiconductor SnO, similar to the low I_OFF_ using ultra-thin body Si-on-Insulator (SOI) and Fin field-effect transistor (FinFET). However, the small sub-10 nm-scale channel thickness can increase the interface roughness scattering and decrease the mobility.

To investigate the mechanism of such annealing temperature dependence, Figure 4a,b plot the I*_DS_* versus drain-source voltage (*I_DS_*-*V_DS_*) and the gate-source current versus gate-source voltage (|*I_GS_*|-*V_GS_*) characteristics of top-gate SnO p-TFTs, respectively, at annealing temperatures of 100 and 200 °C. The p-TFT device at 100 °C annealing shows higher |*I_DS_*| than that at 200 °C annealing, corresponding to the higher *μ_FE_* (Figure 3b). In normal case, a high post-annealing temperature of high-κ gate dielectric is necessary to reduce the gate leakage current and improve the device performance. However, the measured |*I_GS_*| of HfO_2_/SnO p-TFT annealed at 200 °C shows one order of magnitude higher gate leakage than that in the device annealed at 100 °C. The as-deposited HfO_2_ layer without annealing has too high gate leakage current due to defect conduction [41] and unsuitable for device application. To decrease the defect-conductive leakage current, even higher annealing temperature is required for metal-gate/high-κ/Si CMOS [39,42,43].

To further inspect the unusual annealing temperature dependence on device performance, material analysis of atomic force microscope (AFM), X-ray photoelectron spectroscopy (XPS), and secondary ion mass spectrometry (SIMS) were performed. In Appendix A, the surface roughness of the 50 nm HfO_2_ films annealed at 100 °C, 200 °C and 400 °C were analyzed through AFM. The root mean square values of surface roughness show slightly decrease along with the increasement of the annealing temperature. The HfO_2_ dielectric with different annealing temperatures were also analyzed by XPS. As shown in Appendix A, the binding energies of Hf-O and non-lattice O were 530 eV and 531.3 eV, respectively. The peak intensity of non-lattice O was related to the defects in HfO_2_ dielectric, which decreased with increasing post-annealing temperature. From the AFM and XPS analysis, the good device performance at 100 °C annealing is not related to the tiny difference of HfO_2_ layer.

To further investigate the *O_pp_* effect on chemical composition of the SnO layer, the XPS analyses on channel SnO were performed at the *O_pp_* of 14.2%, 25% and 33.3%. The HfO_2_ layer of HfO_2_/SnO stack were etched before the XPS analysis. The XPS data are depicted in Figure 5. The XPS spectra can be deconvolved into three curves from the Sn^2+^O, Sn^4+^O_2_ and Sn^0^ signals, with their corresponding energies of 486.8, 486 and 484.4 eV, respectively. The composition x values of SnO_x_ deposited at *O_pp_* = 14.2%, 25% and 33.3% were 0.8, 0.95 and 1.3, respectively, which explains well the measured electrical data in Figure 2. 

Figure 6a–c show the SIMS profiles of Hf, Sn and O atoms from the HfO_2_/SnO device structure annealed at 100, 200 and 400 °C, respectively. Increasing the annealing temperature from 100 to 400 °C led to significant Sn diffusion from SnO into HfO_2_. This is attributed to the weak Sn–O band enthalpy [38], even though it also leads to high hole mobility [15,16]: SnO → Sn^2+^ + O^2−^(2)

The charged Sn^2+^ can diffuse into HfO_2_, create vacancies at elevated temperatures, and, together with charged O^2−^ ions, allow HfO_x_ diffusion into the SnO layer at 200 °C annealing temperature. The inter-diffusion of Sn and Hf atoms and the formed vacancies and charged ions further degrade the HfO_2_ gate-dielectric and HfO_2_/SnO interface that cause poor |*I_GS_*|, *μ_FE_*, I_ON_, and I_OFF_. The amount of Hf diffusion into the SnO layer at 400 °C annealing temperature can be calculated by the area within SnO layer in Figure 6c, which is 1.15 times higher than the HfO_2_/SnO annealed at 200 °C. The Sn atoms diffused into HfO_2_ layer at 400 °C were 1.14 times more than the HfO_2_/SnO annealed at 200 °C. Thus, the higher post-annealing temperature will cause more inter-diffusion between HfO_2_ and SnO.

The diffused Sn^2+^ can behave as trap states in HfO_2_ gate dielectric, provide extra transport paths for the carriers, and lead to higher gate leakage current (Figure 4b). To understand the conduction mechanism of gate leakage current, the measured data were fitted with various mechanisms. As shown in Figure 7a, the measured |*I_GS_*|-*V_GS_* fits well with the hopping conduction [44,45,46,47], under an electric field (*E*) of <0.25 MV/cm, for 100 and 200 °C annealed top-gate SnO p-TFTs, where the slope of ln(|J*_GS_*|)-*E* is 5.72 and 4.91, respectively. The hopping conduction mechanism is expressed as [45]:(3)$$\left|\;J\;\right|=qan\nu \times exp\left[\frac{qaE}{kT}-\frac{E_{a}}{kT}\right],$$
where J, *q, a*, n, *ν*, and *E_a_* are the current density, electron charge, mean hopping distance, carrier concentration, thermal vibration frequency of carriers at trap states, and activation energy, respectively. The hopping distances of 100 and 200 °C annealed devices calculated from Equation (3) are 1.48 and 1.27 nm, respectively. The smaller hopping distance is ascribed to the Sn diffusion in HfO_2_, which increases the gate leakage *J_GS_*. The mechanism of poor gate leakage current and interface at high annealing temperature is depicted schematically in Figure 7b. The trap-induced hopping conduction causes high |*I_GS_*|. The degraded interface by Sn and Hf inter-diffusion and created vacancies increase the hole scattering from the source to drain, thus lowering the important I_ON_ and *μ_FE_*. The created vacancies also increase the I_OFF_ through defect conduction. The device performance can be further evaluated by the |*I_DS_*|*-V_GS_* hysteresis curves. The defect density formed by hysteresis curves, under forward and reverse sweep between 0 to −3 V, are 1.5 × 10^12^ and 5.4 × 10^12^ cm^−2^ for device annealed at 100 and 200 °C, respectively. This result is consistent to our conclusion: the higher post-annealing temperature creates more defects in the HfO_2_/SnO gate capacitor, which leads to higher gate leakage current, lower hole mobility, and poorer hysteresis than the one annealed at lower 100 °C temperature.

The sub-threshold slope is related to interface trap, which can be calculated [48]:(4)$$\mathrm{SS}=\;\frac{KT}{q}\times ln10\times \;\left(1+\frac{C_{dep}+C_{it}}{C_{ox}}\right)$$
where the *C_dep_*, *C_it_* and *C_ox_* are the depletion capacitance, interface trap capacitance and gate dielectric capacitance, respectively. The interface trap density (D_it_) is 2.5 × 10^13^ eV^−1^cm^−2^ that is higher than the metal-gate/high-κ/Si CMOS. Therefore, the hump of sub-threshold |*I_DS_*|*-V_GS_* curve in Figure 2 is due to the charge modulation from the interface traps [49]. In comparison with SnO atomic density of 2.9 × 10^22^ atoms/cm^3^ or sheet atomic density of 9.4 × 10^14^ atoms/cm^2^, the D_it_ is only 2.7% of the sheet atoms of SnO. Thus, the electronic measurement is highly sensitive to defects compared with other measurements.

The *μ_FE_* data increase to a peak value and decrease with increasing gate field. The detailed physical analysis in oxide semiconductor transistor is not reported yet. However, such hole mobility dependence is generally observed in SiO_2_/Si [50], high-κ/Si [51,52,53], SiO_2_/SiGe [54], high-κ/SiGe [55] and high-κ/Ge [56] p-MOSFETs. Because the Si, SiGe, Ge and SnO are all semiconductors and have the similar valance band structure, the decreased mobility at high electric field may be due to the similar mechanisms of phonon and interface roughness scatterings [57]. 

The La_2_O_3_ can achieve the excellent performance of low leakage current and high-κ value [55,58,59], but the moisture degradation is stronger than the HfO_2_ and ZrO_2_. The ZrO_2_ [60] has a higher κ value than HfO_2_ once crystallized, which is widely used for dynamic random-access memory (DRAM) capacitor. For gate dielectric application, orientation-independent amorphous material like conventional SiO_2_ is needed [61]. The TiO_2_ has the highest κ value but suffers from the small energy bandgap and high leakage current [62]. Thus, the TiO_2_ is generally added to other high-κ dielectric to increase the overall κ value [63]. The Al_2_O_3_ has been used for gate dielectric due to its excellent stability [40], but suffers from relatively lower κ value than HfO_2_. Therefore, the HfO_2_ is used for CMOS application and also for this work.

To inspect the stability of the top-gate SnO TFT devices, the devices were measured at as-fabricated and after retention in air ambient for two months, as depicted in Figure 8. In comparison with the conventional bottom-gate structure, such top-gate device shows huge stability improvement after retention in air [32], which is due to fully covered channel layer by metal-gate and gate-dielectric. Therefore, both the 100 and 200 °C annealed top-gate transistors show only slight degradation after exposure in air for two months. The top-gate HfO_2_/SnO p-TFT has slightly lower hole *μ_FE_* of 4.4 cm^2/^Vs than our previously reported 7.6 cm^2^/Vs of bottom-gate HfO_2_/SnO device, which is attributed to the HfO_2_/SnO inter-diffusion. Because of the larger SS of top-gate device than the bottom-gate one with the same HfO_2_ and SnO, the HfO_2_/SnO interface degradation is confirmed from Equation (4).

Table 1 presents a comparison of the essential device characteristics of top-gate SnO p-TFTs [11,33,34,35] The merits of this work are the highest *μ_FE_* of 4.4 cm^2^/Vs, largest I_ON_/I_OFF_ of 1.2 × 10^5^, and sharpest SS of 526 mV/decade reported to date at fabrication temperatures of only 100–200 °C. This device thus has high potential to be integrated into the IMD layer of Si chips for monolithic 3D and brain-mimicking IC applications.

## 4. Conclusions

While SnO has the advantage of high hole mobility, it also has low bond enthalpy. The key factor for good device integrity of top-gate HfO_2_/SnO p-TFT is to maintain the low process temperature, which can preserve good HfO_2_/SnO interface. Such a low-temperature fabrication (100–200 °C) and excellent device performance are crucial for monolithic 3D and brain-mimicking ICs made on the backend IMD layers of Si chips.

## Figures and Tables

**Figure 1 nanomaterials-11-00092-f001:**
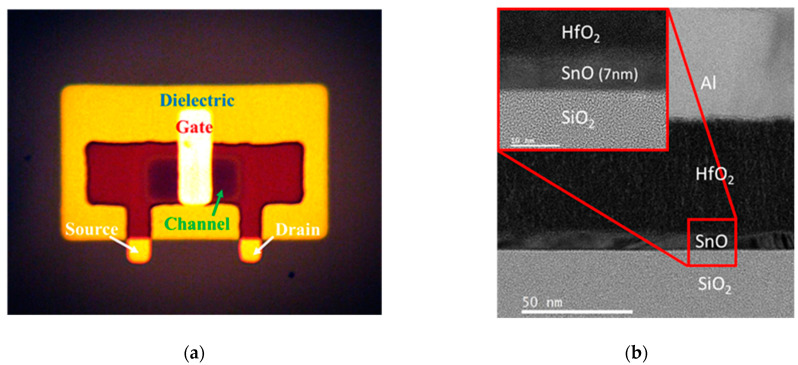
(**a**) top-view photograph and (**b**) cross-sectional TEM image of the top-gate nanosheet SnO p-TFT device. The “white”-color gate on top of the device is due to the light-reflective Al metal.

**Figure 2 nanomaterials-11-00092-f002:**
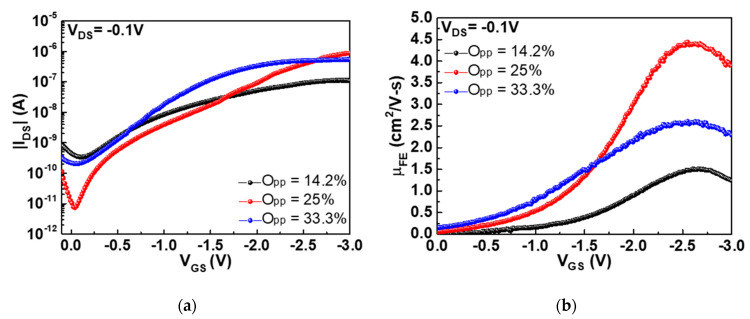
(**a**) |*I_DS_*|*-V_GS_* and (**b**) *μ_FE_-V_GS_* characteristics of top-gate nanosheet SnO p-TFTs with SnO channel deposited at different *O_pp_* conditions. The SnO layer was annealed at 200 °C and HfO_2_ layer was annealed at 100 °C.

**Figure 3 nanomaterials-11-00092-f003:**
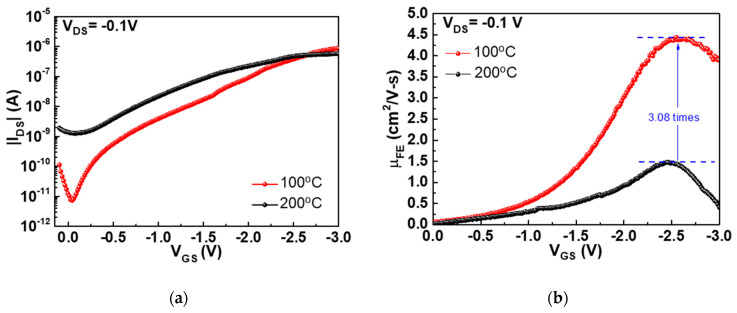
(**a**) |*I_DS_*|*-V_GS_* and (**b**) *μ_FE_-V_GS_* characteristics of top-gate nanosheet SnO TFT annealed at 100 °C N_2_ ambient and 200 °C N_2_ ambient.

**Figure 4 nanomaterials-11-00092-f004:**
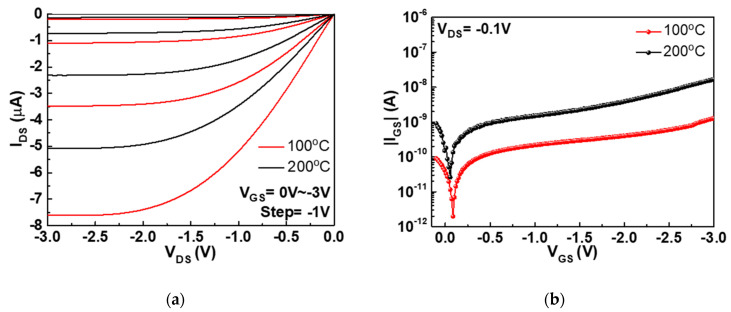
(**a**) *I_DS_-V_DS_* and (**b**) |*I_GS_*|*-V_GS_* characteristics of top-gate nanosheet SnO TFTs annealed at 100 and 200 °C.

**Figure 5 nanomaterials-11-00092-f005:**
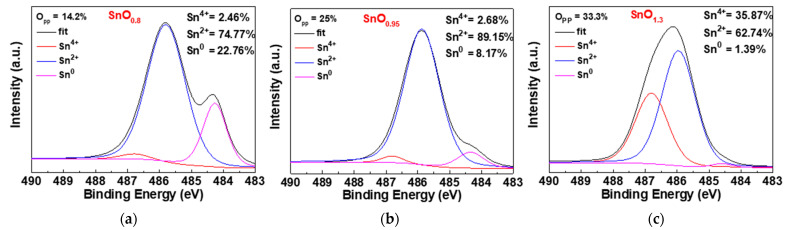
The Sn 3d_5/2_ spectra of SnO films deposited at *O_pp_* of (**a**) 14.2%, (**b**) 25% and (**c**) 33.3%.

**Figure 6 nanomaterials-11-00092-f006:**
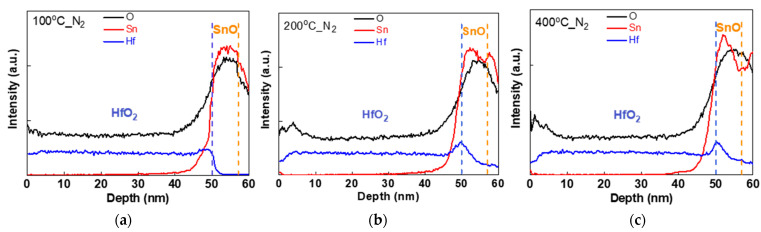
SIMS depth profiles of HfO_2_/SnO stack annealed at (**a**) 100, (**b**) 200 and (**c**) 400 °C N_2_ ambient.

**Figure 7 nanomaterials-11-00092-f007:**
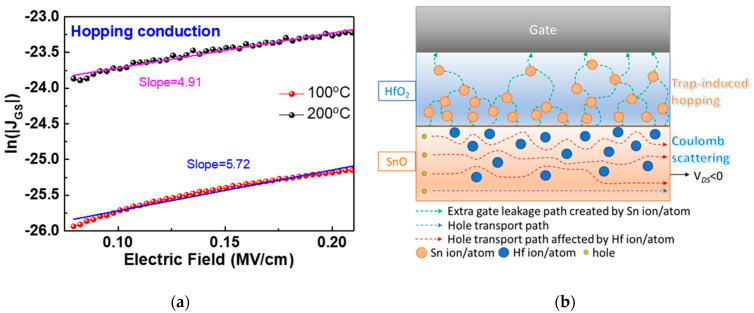
(**a**) |*I_GS_*|-*V_GS_* characteristics of top-gate nanosheet HfO_2_/SnO p-TFTs annealed at 100 and 200 °C. (**b**) Schematic diagram to show gate leakage and inter-diffusion of top-gate SnO TFT device annealed at 200 °C.

**Figure 8 nanomaterials-11-00092-f008:**
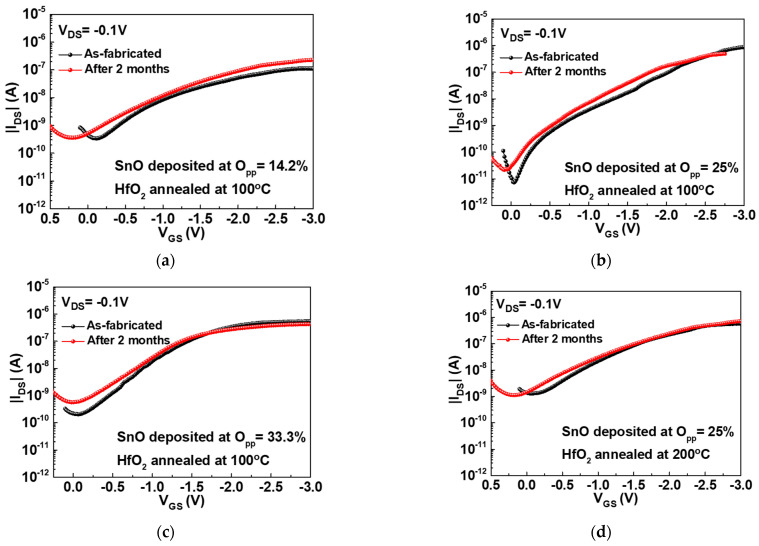
The |I*_DS_*|-*V_GS_* characteristics of the top-gate nanosheet SnO TFT devices with (**a**) SnO deposited at *O_pp_* = 14.2% and HfO_2_ annealed at 100 °C, (**b**) SnO deposited at *O_pp_* = 25% and HfO_2_ annealed at 100 °C, (**c**) SnO deposited at *O_pp_* = 33.3% and HfO_2_ annealed at 100 °C, and (**d**) SnO deposited at *O_pp_* = 25% and HfO_2_ annealed at 200 °C. The devices were measured at as-fabricated and after 2 months’ exposure to air ambient.

**Table 1 nanomaterials-11-00092-t001:** The device performances of various top-gate SnO p-TFTs.

Reference	SnO Thickness (nm)	Gate Insulator Materials	*μ**_FE_* (cm^2^/V·s) @V*_DS_* (V)	I_ON_/I_OFF_	SS (mV/Decade)	Process Temp. (°C)
11	15.4	Y_2_O_3_	0.05 @-1	10^2^	-	250
33	20	Al_2_O_3_	1.3 @-2	10^2^	7	575
34	15	HfO_2_	0.71 @-1	1.6 × 10^3^	1.6	200
35	30	P(VDF-TrFE)	2.7 @-1	2.2 × 10^2^	4	200
This work	7	HfO_2_	4.4 @-0.1	2 × 10^5^	0.526	200

## Data Availability

The data presented in this study are available on request from the corresponding author. The data are not publicly available due to privacy.

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
