# Peer review of "Exceedingly High Performance Top-Gate P-Type SnO Thin Film Transistor with a Nanometer Scale Channel Layer"

_nanomaterials, 2021, doi:10.3390/nano11010092_

Round 1

Reviewer 1 Report

The paper by Yen and coworkers reports the fabrication of a thin film transistor based on SnO in top gate configuration which increase the electrical performance compared withthe more conventional bottom gate configuration. The effect of Oxygen partial pressure for the fabrication of the thin layer of SnO and the influence of the thermal annealing of the dielectric HfO2 on the device performance are presented. A detailed investigation by different techniques are also reported to explain how the farication conditions affect the morphology and the chemical composition of the dielectric and the semiconductor layer, respectively. The authors present clearly the work and this Reviewer suggest the publication of the paper in Nanomaterials Journal after minor revisions.

In figure 2 the transfers and the mobility vs Vgs are reported as function of the Oxygen partial pressure, at which temperature were annealed these devices? For the omaprison with figure 3 it seems these are devices annealed at 100 degree, isn't it? I did not find this info in the main text and maybe it could be added also in the figure caption.

Always regarding the figure 2. The mobility profile if definetely not flat, do the authors know the origin of this? Can the author try to explain this behaviour?

at the line 104 the authors stated that the SnO thinner layer is responsible for the increased Ion/Ioff but at the same time degrades the mobility? Can the author clarify this point?

In figure 5 the XPS spectra are reported but in the main text there isn't a complete description. Can the authors add a small description of these data?

Reviewer 2 Report

Well done work!

Author Response

Thank you for your important comments.

Reviewer 3 Report

In the submitted manuscript, the authors report the fabrication of a top-gate SnO p-TFT with a high μFE of 4.4 cm2/Vs, large ION/IOFF of 1.2×105, and sharp SS of 526 mV/decade. The device exhibits good figure of merits as a remarkably high hole field-effect mobility, sharp SS for low-voltage operation, and a sufficiently large on/off current ratio. The paper resembles previously published work in Ref. [16, 17] and [32].  However, the type of the structure under study is much more challenging for future monolithic 3D-IC applications. Comparative results on the performance metrics of previously fabricated top-gate SnO p-TFTs are also presented by the authors. The paper is well written and the presentation is comprehensive and supported by supplementary experimental material. Therefore, the submitted paper deserves publication.

Author Response

Thank you for your important and constructive comments.

Reviewer 4 Report

This paper demonstrated the effects of oxygen partial pressure during the channel layer deposition and the annealing temperature on the electrical performance of top-gate SnO TFTs. The overall quality is good and the experimental results are interesting. However, there are some questions and suggestions which need to be addressed and clarified before publication.

Q1. Why do the authors use HfO2 as the gate insulator of the fabricated top gate SnO TFT instead of other high-k dielectric materials despite the commented problems ? Do the authors think that the HfO2 is the best material for gate insulator of the SnO TFT ?

Q2. For practical application, not only the electrical performance but the electrical stability is also important. Examine the electrical stability of the fabricated SnO TFTs under each condition and compare the obtained results with the previously reported ones.

Q3. The field-effect mobility of the fabricated TFT decreases with an increase in the gate voltage in Fig. 2(b), which can be attributed to the high contact resistance or the severe surface roughness scattering. It should be relieved. Comment on it in the revised paper.

Q4. There were several previous works on the 3D integration of oxide TFTs. Cite these papers in the introduction part of the revised paper.

Round 2

Reviewer 4 Report

This paper is well revised based on the reviewer's comments and can be published in nanomaterials now.